# Further Characterization of Host Preference of *Acidovorax citrulli* Based on Growth Competition between Group I and Group II Strains

Yuwen Yang [1],*, Pei Qiao [1], Tielin Wang [2], Weiqin Ji [1], Nuoya Fei [3], Liqun Zhang [4], Wei Guan [1] and Tingchang Zhao [1],*

1   State Key Laboratory for Biology of Plant Diseases and Insect Pests, Institute of Plant Protection, Chinese Academy of Agricultural Sciences, Beijing 100193, China
2   State Key Laboratory Breeding Base of Dao-di Herbs, National Resource Center for Chinese Materia Medica, China Academy of Chinese Medical Sciences, Beijing 100700, China
3   Plant Protection College, Shenyang Agricultural University, Shenyang 110866, China
4   College of Plant Protection, China Agricultural University, Beijing 100194, China
*   Correspondence: yangyuwen@caas.cn (Y.Y.); tczhao@ippcaas.cn (T.Z.)

**Abstract:** *Acidovorax citrulli*, the causal agent of bacterial fruit blotch, can be divided into two groups. Group I is mainly isolated from melon, and group II is mainly isolated from watermelon. In this study, group I and II strains were used to assess competition in vivo and in vitro by evaluating inhibition activity assays and by measuring population growth dynamics. Our results indicated that there is no mutual inhibition of growth between the two groups of strains in King's B medium. The bacterial populations of *A. citrulli* strains were counted within 7 days after inoculation in melon and watermelon cotyledons and plotted against time to generate population growth curves. Area under the population growth curves was calculated. When the two groups of strains were inoculated separately into melon and watermelon cotyledons, the results of analysis of variance show that there was no significant difference. In this study, for the first time in an experimental setting, we inoculated two *A. citrulli* groups into melon and watermelon cotyledons at the same time and determined the population dynamics of each strain. The results showed that there was no significant difference between two group strains in melon cotyledons. However, in watermelon cotyledons, the area under population growth curves of group II strain were significantly higher than those of group I strain. Three-way analysis of variance results showed that there was interaction between host and grouping on the influence of strain population level ($p < 0.0001$). These data provide additional information on the host preference of different groups of *A. citrulli*.

**Keywords:** bacterial fruit blotch; growth dynamics; mixed inoculation; melon; watermelon

## 1. Introduction

Bacterial fruit blotch (BFB) caused by *Acidovorax citrulli* is one of the most important seed-borne bacterial diseases on a range of cucurbit crops, such as melon (*Cucumis melo*) and watermelon (*Citrullus lanatus*) [1]. The occurrence of the disease has caused significant economic losses to the melon and watermelon industries in many countries and regions of the world [2–7]. Previous studies have shown that *A. citrulli* can be divided into at least two groups according to their host, multilocus sequence typing (MLST), pulsed field gel electrophoresis (PFGE), *pilA* sequences, DNA fingerprinting, gas chromatography fatty acid methyl ester (GC-FAME), carbon source utilization, and the presence/absence of large DNA fragments [8–11].

Group I strains of *A. citrulli* are mainly isolated from melon and other non-watermelon hosts, and show high levels of virulence to melon and watermelon. On the other hand, group II strains are mainly isolated from watermelon, and the virulence on watermelon is

significantly greater than on other cucurbitaceous species [3,4,12]. Consistent results were obtained by investigating BFB disease incidence in watermelon and melon seedlings after inoculation with different strains [13,14]. Additionally, Yan et al. [15] reported a detached melon fruit assay that clearly indicated host preference of group I and II *A. citrulli* strains. Recently, Zhao et al. [16] showed that different groups of *A. citrulli* strains exhibit host preference under field conditions with natural infection. An important question is "what is the explanation for this phenomenon?" Is it because in commercial cucurbit production a single cultivar of watermelon or melon is usually planted over large areas? In other words, does the absence of mixed cultivation of watermelon and melon reduce the likelihood of cross-infection of the two groups of *A. citrulli* strains in the field? Or is there competition between strains from different groups upon coinfection of a host, leading to one of the strains becoming dominant and surviving locally?

The rapid and accurate detection of group I and II strains serves as the technical basis for investigating in planta competition. For the most part, the detection of *A. citrulli* has focused on discrimination at the species level, and several rapid and sensitive species-specific detection assays have been reported [17–25]. However, recently, the development of rapid detection techniques that can distinguish between *A. citrulli* groups (instar-specific detection assays) has made this current study feasible. In a previous study, we used sequence differences in the type 4 pili (T4P) gene *pilA*, to develop a multiplex PCR-based assay that differentiated different groups of *A. citrulli* strains in one step [11]. Compared with the methods established by Zhong et al. [26] and Zivanovic and Walcott [13], this assay improved the speed and reliability of distinguishing *A. citrulli* strains and can facilitate real-time monitoring of population dynamics in mixed population conditions such as in planta co-inoculation studies.

With the availability of a multiplex PCR assay capable of intraspecific *A. citrulli* discrimination, the objective of this study was to determine if there is competition between group I and II strains during co-culture and co-inoculation. More specifically, our objectives were to (1) determine whether there is competition between group I and II strains, (2) compare the population growth dynamics of group I and II strains when they are cultured individually and in mixed populations in King's B (KB) medium, and (3) compare the population growth dynamics of group I and II strains when they are inoculated individually and together into melon and watermelon cotyledons. Results of this research will help to explain the mechanisms of host preference of strains of the two *A. citrulli* groups on melon and watermelon.

## 2. Materials and Methods

### 2.1. Host Species and Cultivar Culture Conditions

Two-week-old seedlings of melon (*Cucumis melo*) cv. TVF192 and watermelon (*Citrullus lanatus*) cv. Jinxin#3 were used for inoculation with *A. citrulli* [27]. Melon and watermelon seedlings were routinely grown in a growth chamber (Ningbo Jiangnan Instrument Factory, Ningbo, China) with a humidity of 85%, 12 h of light and 12 h of darkness per day, at a temperature of 26 °C during the day and 20 °C at night.

### 2.2. Bacterial Strains and Culture Conditions

*Acidovorax citrulli* group I strains (pslb91, Aacw1, pslb65) and group II strains (pslbtw36, pslb27, Aac5) were used in this study. Group I strain pslb65 was isolated from melon leaves in Xinjiang Uygur Autonomous Region, China; pslb91 strain was isolated from melon fruit in Inner Mongolia Autonomous Region, China; Aacw1 strain was isolated from watermelon seeds in Inner Mongolia, China. Group II strain Aac5 was isolated from watermelon leaves in Taiwan Province, China; pslbtw36 strain was isolated from watermelon fruit in Harbin Province, China; pslb27 strain was isolated from melon leaves in Inner Mongolia, China. *Acidovorax citrulli* strains were grown at 28 °C on KB broth or KA (KB containing 15 g/L agar) media amended with ampicillin at 100 μg/mL [28].

## 2.3. Inhibition Activity Assays between Group I and Group II Strains

Three pairs of strains were used in the test: pslb91 and pslbtw36, Aacw1 and pslb27, pslb65 and Aac5. Each part of the test contains one group I strain and one group II strain. The description of experimental methods is carried out by taking pslb91 and pslbtw36 as examples. The single colony of the two strains was cultured in 5 mL KB liquid medium at 200 r/min 28 °C for 12 h and centrifuged at 3380× *g* for 10 min. Next, the supernatant and cell of the two strains were replaced with each other, that is, the supernatant of pslb91 was injected into the cell of pslbtw36 for resuspension, and the supernatant of pslbtw36 was injected into the cell of pslb91 for resuspension. At the same time, the bacterial suspension that was not exchanged with each other was set as the control. After exchanging the supernatant, the bacterial suspension was cultured for 24 h. Ten microliters solution of gradient dilution to the $10^{-9}$ concentration was plated on KB medium plates and incubated at 28 °C for 36–48 h. Colonies on plates were counted and bacterial population levels were calculated. The experiment was conducted 3 times, each time with 3 replicates. The tests of Aacw1 and pslb27, pslb65 and Aac5 were conducted in the same way.

## 2.4. Competition Assay between Group I and II Strains In Vitro

### 2.4.1. Establishment of the Standard Curve

The strain pslb65 was shake-cultured to $OD_{600} = 1.0$, and the cell suspensions were ten-fold serially diluted to generate a series ranging from $10^{-1}$ to $10^{-8}$. Ten microliters of each bacterial suspension was spread onto KA plates and incubated at 28 °C for 48 h. Each treatment was repeated 18 times. The remaining bacterial suspensions of different concentrations were stored at −80 °C. After 48 h of culture, single colonies on each plate were counted and the original concentrations of the bacterial suspensions were calculated. The populations of cells in the samples stored at −80 °C were estimated using qPCR on an Applied Biosystems 7500 instrument (Applied Biosystems; Waltham, MA, USA), and a standard curve was drawn.

### 2.4.2. Colony Dynamics of the Two Group Strains In Vitro

Single colonies of group I and group II strains were individually inoculated into KB broth supplemented with ampicillin and cultured at 28 °C with shaking (200 r/min) to $3 \times 10^8$ CFU/mL ($OD_{600} = 0.3$). In this experiment, six strains were divided into three pairs: pslb65 and Aac5, pslb91 and pslbtw36, Aacw1 and pslb27. Taking pslb65 and Aac5 as examples, the experimental method is described. The other two pairs of strains were tested in the same method. Twenty mL of pslb65, 20 mL of Aac5, and 20 mL of pslb65 + 20 mL of Aac5 were inoculated into separate 200 mL of KB broth containing ampicillin, and shake-cultured at 28 °C at 200 r/min. Samples (200 µL aliquots) were collected from the cultures at 3 h intervals. A total of ten samples were taken and stored at −80 °C. qPCR was used to determine the cycle threshold (Ct values) of samples of cultures containing pslb65, Aac5, and pslb65 + Aac5 at the different time points. The primers used (pslb65-F/R and AAC00-1-F/R) are shown in Table S1. The qPCR procedure was conducted as described in the instructions for SuperReal PreMix Plus (SYBR Green) (Tiangen; Beijing, China). The experiment was conducted three times, each time with three replicates. We used the obtained Ct value in the standard curve growth equation to extrapolate the corresponding bacterial concentration of each cell suspension. *A. citrulli* colony dynamics were plotted against time to generate population growth curves. Area under population growth curves (AUPGC) data were calculated and used to compare the population growth dynamics of two strains in difference culture conditions.

## 2.5. Competition Assay between Group I and II Strains in Melon and Watermelon Cotyledons

### 2.5.1. Establishment of the Standard Curve for Determination of Colony Number in Melon and Watermelon

Positive control plasmid construction. Partial sequences of the *pilA* gene from *A. citrulli* strains pslb65 and Aac5 were amplified using their respective specific primers (Table S1).

The plasmid pk18mob*sacB* was digested with *Hind*III and *Eco*RI. The seamless ligase (ClonExpress II One Step Cloning Kit, Vazyme Biotech; Nanjing, China) was used to ligate the partial *pilA* genes from pslb65 and Aac5 to the digested pk18 plasmid individually. Thereafter, the two ligation products were transferred into *Escherichia coli* DH5α. The pk18-pslb65*pilA* and pk18-A5*pilA* plasmids were extracted from positive transformants, which were verified by PCR and sequencing, and stored at −20 °C after the concentrations were determined.

### 2.5.2. Establishment of the Standard Curve

The plasmids were 10-fold serial diluted with DEPC water, and eight 10-fold serial dilutions ($10^0 - 10^{-7}$) were used as templates for qPCR. The qPCR reaction was carried out using the SuperReal PreMix Plus (Tiangen; Beijing, China). Three technical repeats were made for each concentration gradient. When the qPCR reaction was completed, Excel was used to plot the logarithm of the template concentration as the abscissa and the Ct value was used as the ordinate to generate the standard curve. In order to make accurate comparisons between different samples at different time points, the concentration of DNA was converted into a template copy number (CN), as follows: CN = (M × N)/(L × D), where M = DNA concentration (g/mL) = DNA concentration (ng/L) × $10^{-6}$, N = Avogadro's constant ($6.022 \times 10^{23}$ molecule/mol), L = Nucleic acid molecule length (Total length (kb) = target fragment + vector), D = Conversion factor (dsDNA conversion rate is $6.6 \times 10^5$ g/(mol·kb)). The CN/Total DNA Content was used as the bacterial content unit.

### 2.5.3. Colony Dynamics of Group I and II Strains in Melon and Watermelon Cotyledons

The test was divided into three parts: pslb65 and Aac5, pslb91 and pslbtw36, Aacw1 and pslb27. The experimental method is described with pslb65 and Aac5 as examples. *Acidovorax citrulli* strains pslb65 and Aac5 were cultured overnight in KB broth, adjusted to an $OD_{600} = 0.3$ (approximately $3 \times 10^8$ CFU/mL), and then ten-fold serially diluted to $3 \times 10^4$ CFU/mL. The cell suspensions of pslb65, Aac5 and 1:1 mixed suspension of pslb65 and Aac5 were injected into two-week-old watermelon and melon cotyledons to fill the cotyledons respectively. As a negative control, plants were injected with sterilized water. Beginning the day after inoculation, four treatment sets of melon and watermelon seedlings were established (pslb65 alone, Aac5 alone, pslb65 and Aac5 mixed, and sterilized water). Every 24 h after inoculation, four cotyledons from two seedlings were sampled from each treatment set. The cotyledons were weighed and placed into a centrifuge tube for pulverization using the FastPrep-24 5G Sample Preparation Instrument (MP Biomedicals; Santa Ana, California, USA). The total DNA from each treatment was extracted using the Multisource Genomic DNA Miniprep Kit (Axygen; Hangzhou, China). After the DNA concentration was standardized using the NanoVue Plus (GE, Biochrom Ltd.; Boston, MA, USA), qPCR was used to quantify the populations of pslb65 and Aac5 on watermelon and melon cotyledons [29]. *Acidovorax citrulli* colony dynamics were plotted against time to generate population growth curves. AUPGC were calculated and used to compare the population growth dynamics of two strains in difference hosts. The melon *CmACT* gene was used as a reference gene in the melon test [30]. The watermelon *ClTUA* gene was used as a reference gene in the watermelon test [31]. Relative expression of *pilA* genes was calculated by the $2^{-\Delta\Delta CT}$ method [32]. The primers used in this study were synthesized by BGI Laboratories (BGI Group, Shenzhen, China). The primers used for qPCR are listed in Table S1. Each sample was assayed in triplicate.

### 2.6. Data Processing/Statistical Analysis

The experimental data were recorded and calculated using Excel 2016 (Microsoft; Redmond, WA, USA), and graphed using GraphPad Prism 7 (GraphPad; San Diego, CA, USA). The growth inhibition test of two groups of strains was conducted by independent sample *t*-test. Analysis of variance (ANOVA) was conducted to determine the significance of the effect of strain, least significant difference tests were conducted to compare the

effects of *A. citrulli* strain on bacterial colonization of host cotyledons. In order to explore the interaction between host and strain grouping, three-way ANOVA was performed on pooled date from three replicates of the experiment. All analyses were conducted in IBM SPSS Statistics version 22 (IBM SPSS Inc., Chicago, IL, USA).

## 3. Results

### 3.1. Cell-Free Culture Supernatants from A. Citrulli Group I or Group II Strains Do Not Inhibit Each Other's Growth in Culture

The results of colony population statistics showed that in the first pair of strains, there was no significant difference ($p = 0.528$) in the colony population of group I strain pslb91 between the supernatant of group I strain pslbt91 and group II strain pslbtw36, and there was also no significant difference ($p = 0.202$) in the colony population of group II strain pslbtw36 between these two types of supernatants. The test results in the second pair of strains were similar to those of the first pair of strains, the growth population of AacW1 in the supernatants of AacW1 and pslb27 had no significant difference ($p = 0.516$), and the growth population of pslb27 in the supernatants of AacW1 and pslb27 had no significant difference ($p = 0.581$). The test results in the third pair of strains were similar to those of the first and the second pairs, the growth population of Aac5 in the supernatants of Aac5 and pslb65 had no significant difference ($p = 0.783$) and the growth population of pslb65 in the supernatants of Aac5 and pslb65 had no significant difference ($p = 0.594$) (Figure 1).

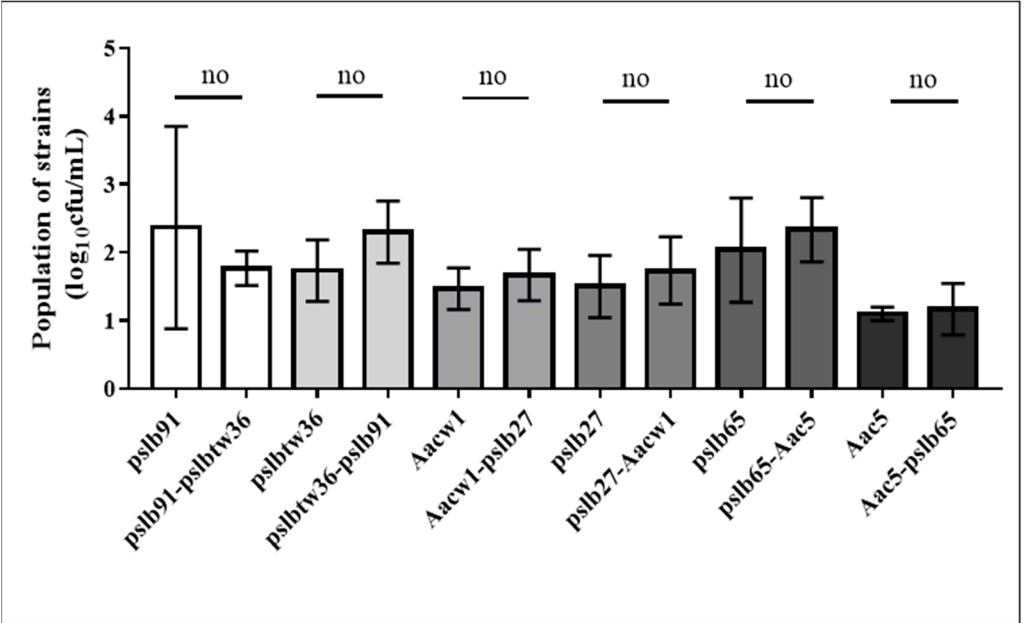

**Figure 1.** Growth inhibition assay of *Acidovorax citrulli* group I and group II strains. Three pairs of *A. citrulli* strains were used: pslb91 and pslbtw36, Aacw1 and pslb27, pslb65 and Aac5. Among them, pslb91, Aacw1, and pslb65 belong to group I, and pslbtw36, pslb27, and Aac5 belong to group II. In the figure, pslb91 refer to the bacterial population of pslb91 strain respectively after 36 h of shaking culture in KB liquid medium at 28 °C, pslbtw36 refer to that of pslbtw36 strain. The supernatant and cell of the two strains cultured by shaking for 12 h were replaced with each other, that is, the supernatant of pslb91 was injected into the cell of pslbtw36 for resuspension, recorded as pslbtw36-pslb91, and the supernatant of pslbtw36 was injected into the cell of pslb91 for resuspension, recorded as pslbtw36-pslb91. The bacterial population was counted after 24 h of culture. The other two pairs of strains are the same. The *t*-test was used to calculate the difference in the population of strains in different treatments, and "no" above the bar chart indicates no significant difference ($p > 0.05$).

### 3.2. Colony Dynamics of the Two Group Strains When Cultured Separately In Vitro

The standard curve for converting the cycle threshold (Ct) value into citrulline eosinophil cell concentration using quantitative real-time PCR (qPCR) is shown in Figure S1. qPCR results showed that when group I strains (pslb65, pslb91 and Aacw1) and group II strains (Aac5, pslbtw36, pslb27) were cultured in vitro separately, the six strains showed similar growth curves (Figure 2a). Based on the AUPGC, the differences in population growth of the six strains were not significant ($p$ = 0.131) (Figure 2b).

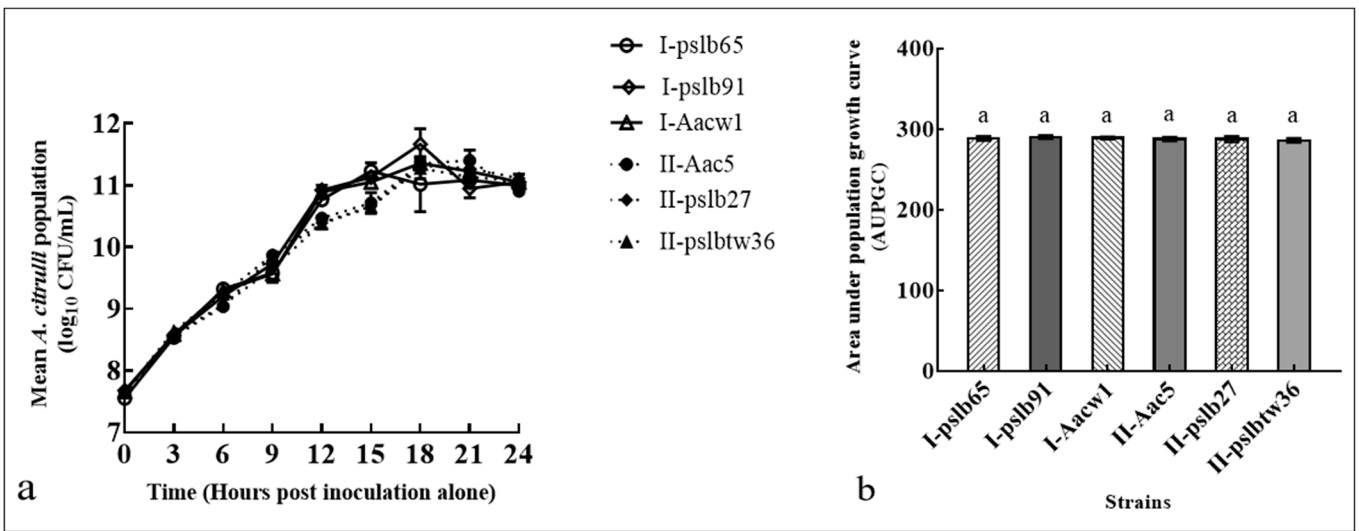

**Figure 2.** Population dynamics of *Acidovorax citrulli* strains cultured separately in King's B medium. (**a**) 20 mL aliquots of cell suspensions ($10^8$ CFU/mL) of the six representative strains were inoculated into 200 mL KB broth, and shake-cultured at 28 °C at 200 r/min. 200 μL samples were taken every 3 h. qPCR was used to determine the cycle threshold (Ct) values for each strain at different time points. (**b**) Bar chart of area under population growth curves calculated for each strain cultured individually in KB medium. There were no significant differences between same letter substitutes on the column calculated by Duncan's Multiple Range Test $p$ = 0.05.

### 3.3. Colony Dynamics of Group I or Group II Strains in Mixed Culture In Vitro

The same number of three pairs of strains (pslb91 and pslbtw36, Aacw1 and pslb27, pslb65 and Aac5) cells were simultaneously inoculated into KB medium, and the respective colony dynamics were determined. The data indicated that the number of group I strain colony forming units (CFU) was slightly higher than that of group II strains of each pair in logarithmic phase. When cultured for 18 h, the strains reached the logarithmic phase, and the number of colonies of group I and group II strains in each pair of strains was basically the same, about $10^{11}$ CFU/mL (Figure 3a–c). Based on the AUPGC, the differences in population growth of the three pairs of strains were not significant. The significance of pslb91 and pslbtw36 was $p$ = 0.120, that of Aacw1 and pslb27 was $p$ = 0.555, that of pslb65 and Aac5 was $p$ = 0.804 (Figure 3d).

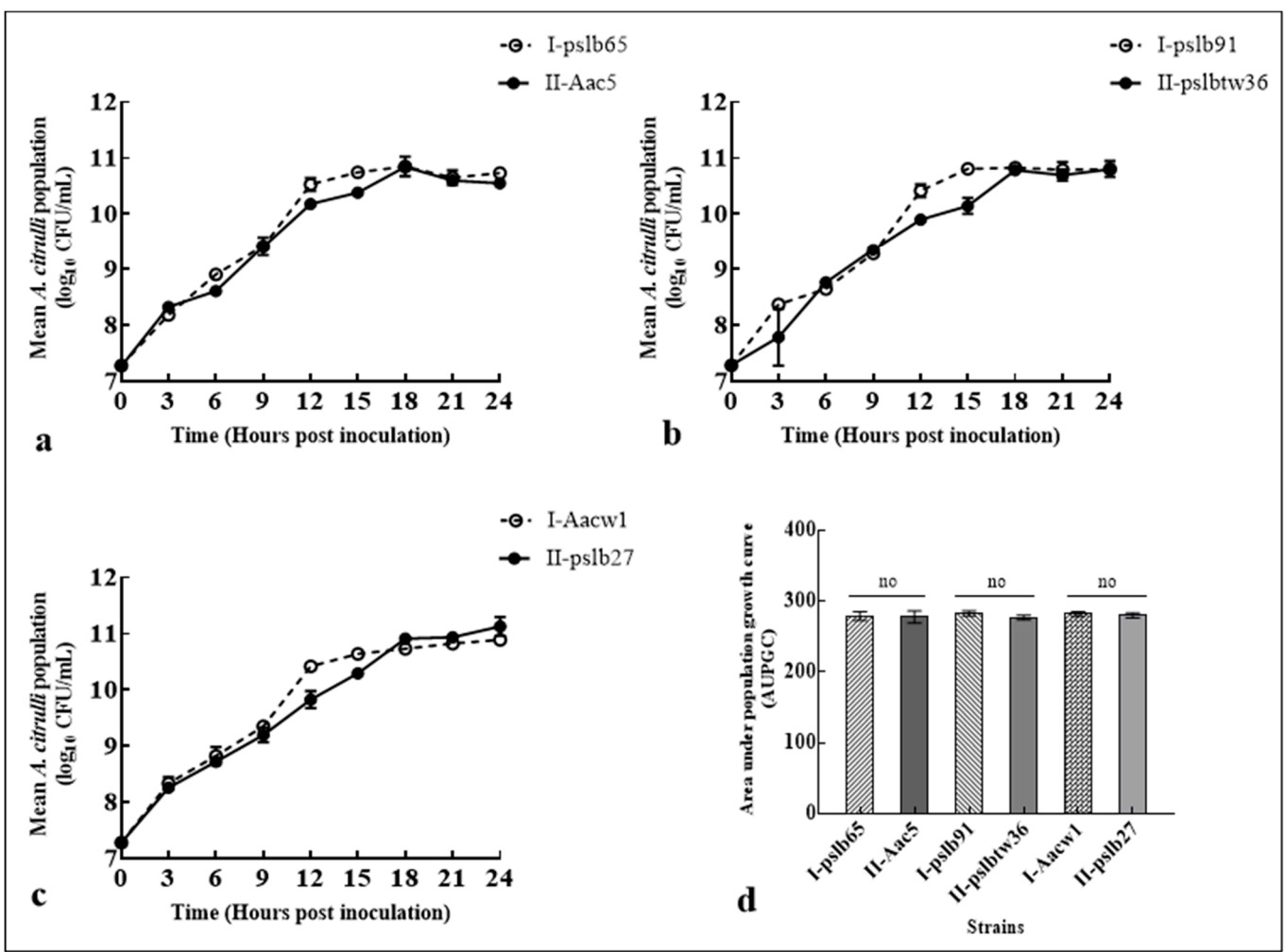

**Figure 3.** Population dynamics of *Acidovorax citrulli* group I and group II strains co-cultured in King's B medium. (**a**) Group I pslb65 strain and group II Aac5 strain were mixed in equal proportion and cultured in the KB medium. Aliquots (20 mL) of cell suspensions ($10^8$ CFU/mL) of pslb65 and Aac5 were inoculated into 200 mL KB broth together, and shake-cultured at 28 °C at 200 r/min. Every 3 h, 200 μL samples were taken. The qPCR technique was used to determine the Ct values of pslb65 and Aac5 populations at different time points. (**b**) Group I pslb91 strain and group II pslbtw36 strain were mixed in equal proportion and cultured in the KB medium. (**c**) Group I Aacw1 strain and group II pslb27 strain were mixed in equal proportion and cultured in the KB medium. (**d**) Bar chart of AUPGC calculated for group I and group II strains co-cultured in equal proportion in KB medium. The experiment was conducted twice and each treatment replicated three times. Bars represent mean AUPGC values, lines represent standard errors. The *t*-test was used to calculate the difference of AUPGC between group I and group II strains, and "no" above the bar chart indicates no significant difference (*p* = 0.05).

### 3.4. Colony Dynamics of Group I and Group II Strains Individually Inoculated into Melon and Watermelon Cotyledons

The standard curve for converting cycle threshold (Ct) values to the concentrations of *A. citrulli* strains pslb65 and Aac5 using quantitative real-time PCR are shown in Figure S2. After the injection of melon cotyledons, the colony growth dynamic curves of the six strains were drawn according to their respective population numbers. The results showed that the six strains had similar dynamic curves, and the maximum colony numbers were all within $10^8$ and $10^9$ CFU/ng (Figure 4a). Based on the AUPGC, T test results of independent samples showed that the differences between the six strains population growth were significant in melon cotyledons. The AUPGC of Aac5 strains was significantly

($p < 0.05$) lower than that of pslb91, pslbtw36 and Aacw1, no significant difference with pslb27 and pslb65 (Figure 4b). However, the difference between group I strains (pslb91, Aacw1 and pslb65) and group II strains (pslbtw36, pslb27 and Aac5) ($p > 0.05$) were not significant by single factor analysis of variance.

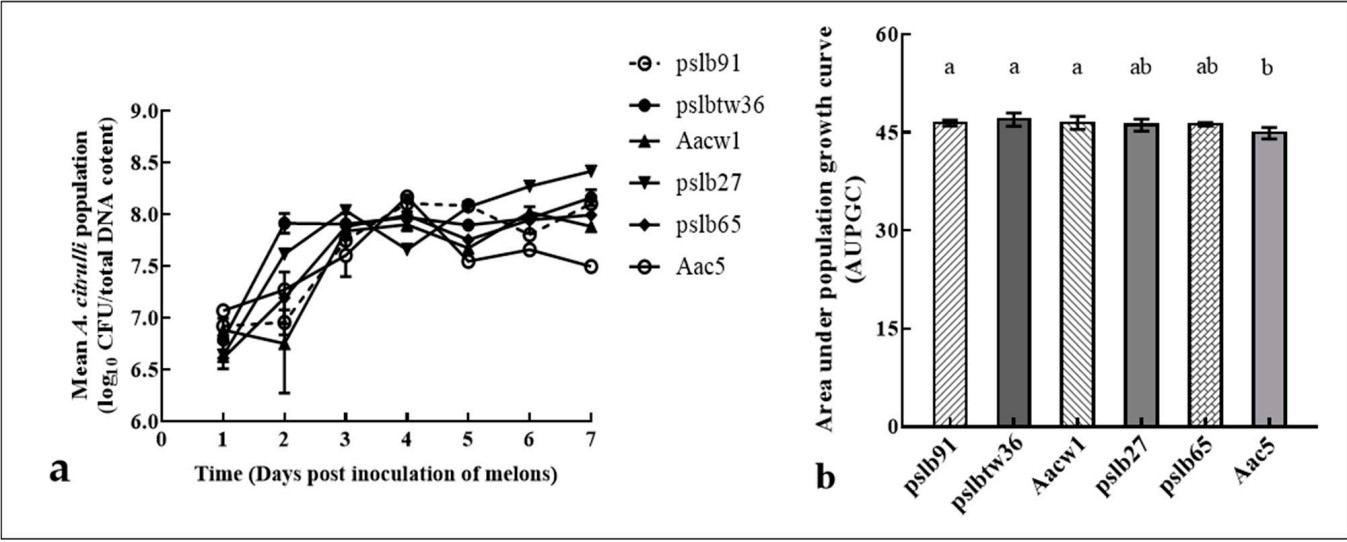

**Figure 4.** Temporal population dynamics after inoculation of *Acidovorax citrulli* group I and group II strains, separately, into melon cotyledons. (**a**) Melon cotyledons after injection inoculation of the six strains. (**b**) Bar chart of area under population growth curves calculated for the six strains in melon. $3 \times 10^4$ CFU/mL group I and II strains were injected into the underside of two-week-old melon cotyledons, sterilized water was used as negative control. From the day following inoculation, two melon seedlings were taken daily for each treatment. Total DNA were separately extracted in each set of treatment. After the DNA concentration was standardized, qPCR was used to quantitatively determine the population dynamics of group I and II strains in melon cotyledons. The experiment was conducted twice and each treatment replicated three times. Bars represent mean AUPGC values, lines represent standard errors, and the different letters on the bar chart indicate significant differences in AUPGC between different strains calculated by ANOVA (Duncan, $p = 0.05$).

After the injection of watermelon cotyledons, the colony growth dynamics of the six strains also showed similar dynamic curve shape as the results of inoculating melon cotyledons. The maximum colony number of each strain was also within $10^8$ and $10^9$ CFU/ng (Figure 5a). Based on the AUPGC, there was no significant difference between the six strains (Figure 5b).

The AUPGC values of the same strain inoculated with different hosts were statistically analyzed. The results showed that each of the six strains had no significant difference when inoculated with melon and watermelon cotyledons.

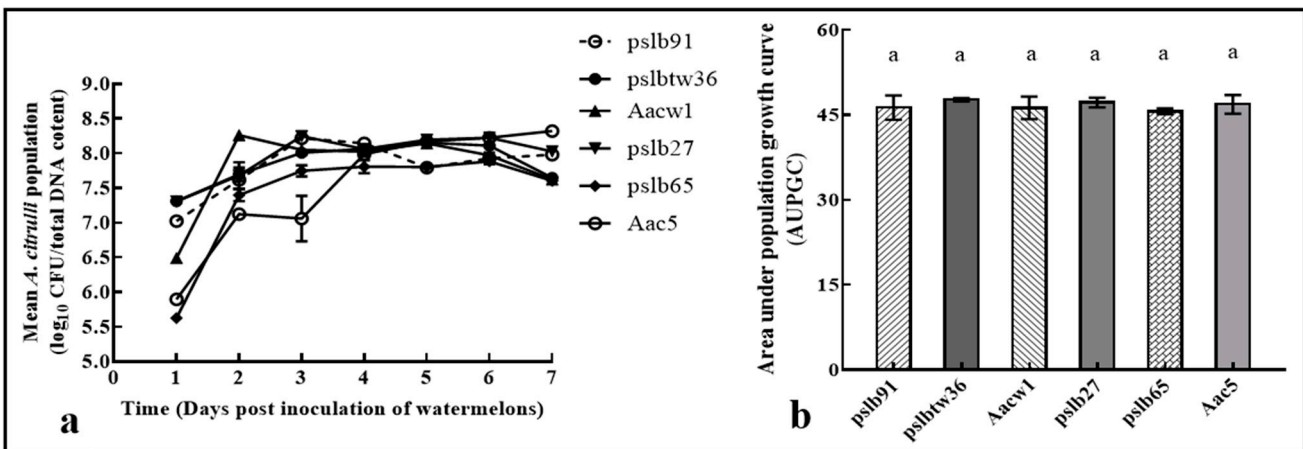

**Figure 5.** Temporal population dynamics after inoculation of *Acidovorax citrulli* group I and group II strains, separately, into watermelon cotyledons. (**a**) Watermelon cotyledons after injection inoculation of the six strains. (**b**) Bar chart of area under population growth curves calculated for the six strains in watermelon. Group I and II strains ($3 \times 10^4$ CFU/mL) were injected into the underside of two-week-old watermelon cotyledons, sterilized water was used as negative control. From the day following inoculation, two watermelon seedlings were taken daily for each treatment. Total DNA were separately extracted in each set of treatments. After the DNA concentration was standardized, qPCR was used to quantitatively determine the population dynamics of group I and II strains in watermelon cotyledons. The experiment was conducted twice and each treatment replicated three times. Bars represent mean AUPGC values, lines represent standard errors, and the different letters on the bar chart indicate significant differences in AUPGC between different strains calculated by ANOVA (Duncan, $p = 0.05$).

*3.5. Colony Dynamics of Group I and Group II Strains after Mixed Inoculation into Melon and Watermelon Cotyledons*

After the three pairs of strains were co-inoculated into melon cotyledons respectively, the maximum colony number of strains in group I and group II was different. The maximum colony number of group I strains (pslb91, Aacw1, pslb65) in each pair was higher than $10^8$ CFU/mL, while group II strains had different situations. Of group II strains, only the population of pslbtw36 was higher than $10^8$ CFU/mL at six to seven days after inoculation (Figure 6a), while the population of pslb27 reached the maximum value of $6.0 \times 10^7$ CFU/mL on the fifth day after inoculation (Figure 6b), Aac5 population reached the maximum value on the second day after inoculation, which was at $5.0 \times 10^6$ CFU/mL, and then the number of colonies gradually stabilized at this level (Figure 6c). Based on the statistical analysis of the AUPGC in melon cotyledons, population growth of pslb65 was significantly ($p < 0.05$) higher than that of Aac5. However, there was no significant difference between the other two pairs of strains (pslb91 and pslbtw36, Aacw1 and pslb27) (Figure 6d).

Compared with the melon experiments, in each strain pair, the group I strain population was much higher than that of the group II strain in watermelon cotyledons after mixed inoculation. In the first pair of strains, the population of group II strain pslbtw36 was higher than that of group I strain pslb91 at one to seven DAI, and the maximum population of pslbtw36 is $1 \times 10^8$ CFU/mL, while the maximum pslb91 population is only $5 \times 10^7$ CFU/mL (Figure 7a). The population growth dynamics of the second pair of strains (Aacw1 and pslb27) are basically the same as those of the first pair (Figure 7b). Within one to three DAI, the group I strain pslb65 population was slightly higher than the group II strain Aac5 population in the third pair of strains. However, at four to seven DAI, the group II strain Aac5 population increased rapidly, reaching $1 \times 10^8$ CFU/mL, while the population of group I strain pslb65 grew slowly, the maximum population was only $8 \times 10^6$ CFU/mL (Figure 7c). Based on the statistical analysis of AUPGC in watermelon

cotyledons, the population growth of group II strains (pslbtw36, pslb27 and Aac5) was significantly higher ($p < 0.05$) than that of group I strains (pslb91, Aacw1, pslb65) in three pairs of strains respectively (Figure 7d).

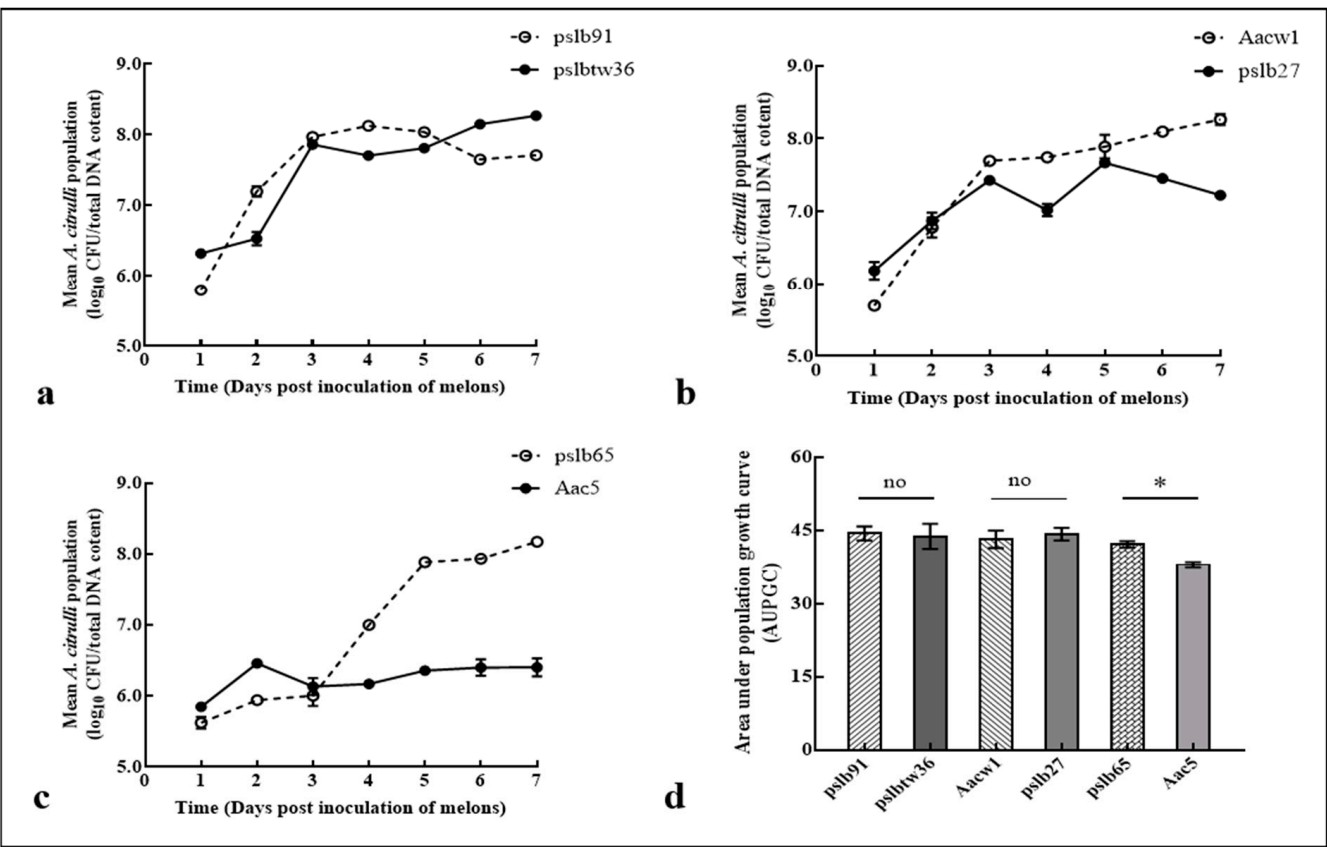

**Figure 6.** Temporal population dynamics of *Acidovorax citrulli* group I and II strains after co-inoculation into melon cotyledons. (**a**) Melon cotyledons after 1:1 mixed injection inoculation of pslb91 and pslbtw36 strains. (**b**) Melon cotyledons after 1:1 mixed injection inoculation of Aacw1 and pslb27 strains. (**c**) Melon cotyledons after 1:1 mixed injection inoculation of pslb65 and Aac5 strains. (**d**) Bar chart of AUPGC calculated for group I and group II strains in melon. A 1:1 mixture ($3 \times 10^4$ CFU/mL group I and group II strains) of cells was injected into the underside of two-week-old melon cotyledons. Each 24 h beginning with the day following inoculation, two melon seedlings were taken for total DNA extraction. After standardizing the DNA concentration, qPCR was used to quantitatively determine the population dynamics of group I and group II strains on melon cotyledons. The experiment was conducted twice and each treatment replicated three times. Bars represent mean AUPGC values, lines represent standard errors. The *t*-test was used to calculate the difference of AUPGC between group I and group II strains, "*" above the bar chart indicates significant difference, and "no" above the bar chart indicates no significant difference ($p = 0.05$).

Three-way ANOVA results showed that there were significant differences in AUPGC between group I and group II strains ($p = 0.0199$), significant differences between melon and watermelon ($p = 0.042$), and extremely significant differences among the six strains ($p < 0.0001$); there was interaction between host and grouping on the influence of strain population level ($p < 0.0001$) (Table S2).

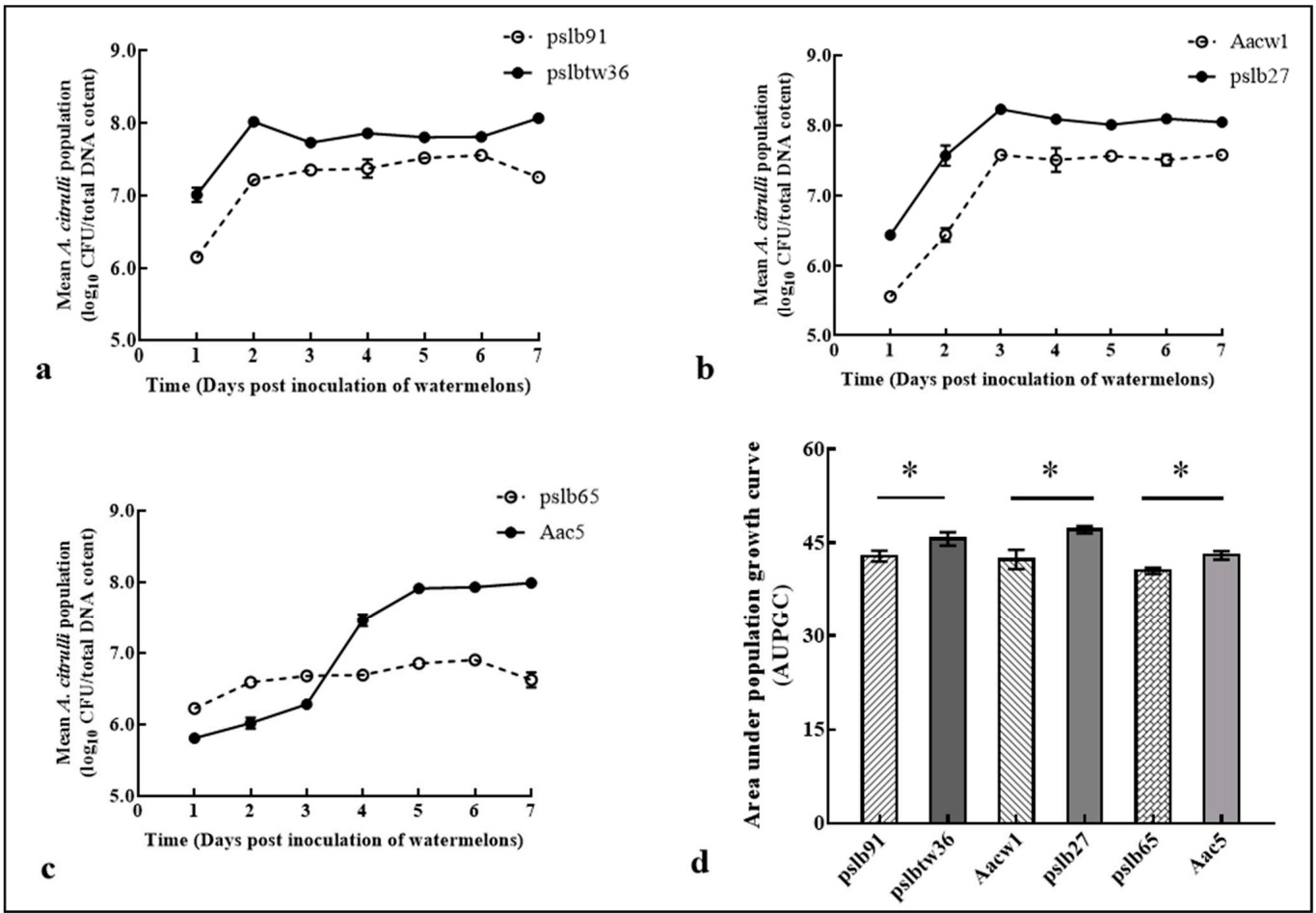

**Figure 7.** Temporal population dynamics of *Acidovorax citrulli* group I and II strains after co-inoculation into watermelon cotyledons. (**a**) Watermelon cotyledons after 1:1 mixed injection inoculation of pslb91 and pslbtw36 strains. (**b**) Watermelon cotyledons after 1:1 mixed injection inoculation of Aacw1 and pslb27 strains. (**c**) Watermelon cotyledons after 1:1 mixed injection inoculation of pslb65 and Aac5 strains. (**d**) Bar chart of AUPGC calculated for group I and group II strains in watermelon. A 1:1 mixture ($3 \times 10^4$ CFU/mL group I and group II strains) of cells was injected into the underside of two-week-old watermelon cotyledons. Each 24 h beginning with the day following inoculation, two watermelon seedlings were taken for total DNA extraction. After standardizing the DNA concentration, qPCR was used to quantitatively determine the population dynamics of group I and group II strains on watermelon cotyledons. The experiment was conducted twice and each treatment replicated three times. Bars represent mean AUPGC values, lines represent standard errors. The *t*-test was used to calculate the difference of AUPGC between group I and group II strains, "*" above the bar chart indicates significant difference, and "no" above the bar chart indicates no significant difference ($p = 0.05$).

## 4. Discussion

There is abundant intraspecific genetic diversity in *A. citrulli*, which can be divided into two groups according to host specificity [3]. In this study, group I (pslb91, Aacw1, pslb65) and group II (pslbtw36, pslb27 and Aac5) strains were used to study inter-group growth inhibition and growth competition in KB medium and different hosts (melon and watermelon). The results provide more evidence for host preference in *A. citrulli*.

Here, we found that there was no growth inhibition between group I and group II strains in the mixed culture in KB medium. Inhibition activity assays revealed that the cell-free supernatant of group I strains (pslb91, Aacw1, pslb65) cannot inhibit the population of group II strains (pslbtw36, pslb27, Aac5) separately. Similarly, the cell-free supernatant

of group II strains (pslbtw36, pslb27, Aac5) also cannot inhibit the population of group I strains (pslb91, Aacw1, pslb65) separately. The population dynamics of group I and group II strains inoculated separately into melon and watermelon cotyledons in our study showed no significant difference, similar to those reported by Zivanovic and Walcott [13]. However, when the melon cotyledons were inoculated, the AUPGC values of the six strains were significantly different which was somewhat different from that when the watermelon cotyledons were inoculated. The AUPGC of Aac5 strain was significantly lower than that of pslb91, pslbtw36 and Aacw1 when the melon cotyledons were injected, while there was no significant difference among the other five strains, indicating that there were individual differences among the three group II strains selected in this study. This study reported for the first time the results of inoculating melon and watermelon cotyledons simultaneously with strains of two *A. citrulli* groups. The results showed that there was no significant difference in colony level between the two groups of strains in melon, but significant difference in watermelon. At the same time, three-way ANOVA showed that there was interaction between host and strain groups. When group I and group II strains were co-inoculated into melon cotyledons, there was no significant difference in the AUPGC values between the two groups of strains in two of the three pairs of mixed strains, and there was significant difference in the AUPGC values between one pair of mixed strains (pslb65 and Aac5). We found that the AUPGC value of Aac5 was the lowest and significantly lower than that of the other three strains when the six strains were inoculated into melon cotyledons individually. It can be seen that the growth ability of Aac5 strain in melon cotyledons is weak, which may be the reason why the AUPGC of Aac5 is significantly lower than that of pslb65 when the melon was co-inoculated. In this study, the inoculation method of *A. citrulli* was cotyledon injection, and the experimental results well reflected the preference of the two group strains in different host cotyledons. In addition, the experiment of co-inoculation on watermelon and melon fruit should also be carried out to further verify the growth competition differences of different groups strains.

Combining the results of the inhibition activity assay and the in vitro and in vivo growth competition assays for two group strains, we speculate that the differences in growth of the two group strains inoculated into different hosts was caused by the complex interaction between the strains and hosts. In our previous study [14], the transcriptomes of melon and watermelon seedlings spray inoculated with pslb65 (*A. citrulli* group I strain) and AAC00-1 (*A. citrulli* group II strain) were assessed at different time points (0 h, 6 h, 12 h). Transcriptome data showed that there are some differences in the plant–pathogen interaction pathways between melon-pslb65 and melon-AAC00-1, and between watermelon-pslb65 and watermelon-AAC00-1 combinations. There were also significant differences in gene regulation between melon and watermelon at different time points after inoculation with the different strains [15]. This suggests that the response mechanisms of group I and group II strains to watermelon and melon are different.

Complex genetic interactions are the basis for host range evolution in plant pathogens. Even when the interaction between a host and its adaptive pathogens shows only a relatively small difference, there may be a complex genetic basis among the pathogens [33]. There are substantial genomic differences between the group I and group II *A. citrulli* strains [8]. The whole genome sequence of the group I strain M6 is about 500 kb shorter than that of the group II strain AAC00-1. The difference is mainly concentrated in eight fragments, which are absent in the genome of the group I strains. Eckshtain-Levi et al. [34] conducted a comparative analysis of eleven effector genes from 22 *A. citrulli* strains, and the results confirmed the differences in type III effectors Aave_2708, Aave_2166 and Aave_3062 between group I and group II strains. The host preference to immature fruit tissues between group I and II *A. citrulli* strains was observed. Typical water-soaked lesions appeared on melon fruit inoculated with group I strain at 7 to 10 DAI. In contrast, during the same times, similar symptoms were not found on melon fruit inoculated with group II strain. The *hrcV* (Type III effector gene) deletion mutants of *A. citrulli* group I strain M6 lost the ability to induce water soaked lesions in melon fruit [15]. It is believed that type III effectors are related to the

pathoadaptive evolution of plant pathogenic *Acidovorax* species [35]. These genetic differences in type III secreted effectors may be one of the reasons for the differences in host adaptability in the two groups of strains. Whether this is indeed the case requires further investigation.

## 5. Conclusions

In this study, representative group I and II strains, respectively, were used to assess competition in vivo and in vitro. The results showed that there was no significant difference in strain population level between the two groups of strains inoculated separately on melon and watermelon cotyledons; when the two groups of strains were co-inoculated into melon and watermelon in equal proportion at the same time, there was no significant difference in the population level of melon cotyledons, but there was significant difference in watermelon cotyledons. It shows that the group II strains have preference for watermelon.

**Supplementary Materials:** The following supporting information can be downloaded at: https://www.mdpi.com/article/10.3390/horticulturae8121173/s1, Table S1: Information about primers used for qPCR in this study; Table S2: Results of three-way ANOVA; Figure S1: The standard curve for converting cycle threshold (Ct) values to the concentration of *Acidovorax citrulli* cells, using quantitative real-time PCR (qPCR); Figure S2: The standard curve for converting cycle threshold (Ct) values to the concentrations of *Acidovorax citrulli* strains pslb65 and Aac5 using quantitative real-time PCR.

**Author Contributions:** T.Z. and Y.Y. designed the study. Y.Y., P.Q., N.F. and W.J. performed the experiments. T.W., Y.Y. and W.G. performed data analyses. Y.Y. wrote the manuscript. T.Z. and L.Z. critically reviewed the manuscript. All authors have read and agreed to the published version of the manuscript.

**Funding:** This study was supported by the China Earmarked Fund for Modern Agro-industry Technology Research System (CARS-25), the Key Project at Central Government Level: The Ability Establishment of Sustainable Use for Valuable Chinese Medicine Resources (2060302), and the Agricultural Science and Technology Innovation Program of the Chinese Academy of Agricultural Sciences (CAAS-ASTIP).

**Institutional Review Board Statement:** Not applicable.

**Informed Consent Statement:** Not applicable.

**Data Availability Statement:** Not applicable.

**Conflicts of Interest:** The authors declare no conflict of interest.

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
