# Peer review of "Further Characterization of Host Preference of Acidovorax citrulli Based on Growth Competition between Group I and Group II Strains"

_horticulturae, doi:10.3390/horticulturae8121173_

Round 1
Reviewer 1 Report
Further characterization of host preference of Acidovorax citrulli based on growth competition between group I and group II strains
The manuscript discusses the host preference of A. citrulli (melon or watermelon) and if there is competition between group 1 and group 2 if inoculated at the same time on cotyledons. They also examine if there are any antagonistic produces by growing each group in the other groups filtrate.
In general, the manuscript is scientifically sound however there are some minor issues throughout and most of the discussion seems to center on a previous genomic study. The manuscript discussion section would be improved by expanding the discussion around the results of this study and limiting the genomic analysis, which was not part of this study or study results. That does not mean some genomic interpretation is not warranted but limit it to how it directly relates to these results.
Abstract suggestions: Typically abstracts should be free of abbreviations and abbreviations should start in the introduction. For example, there is no need to have BFB, DAI, AUPGC in the abstract. Move the first abbreviation to the first mention in the main text.
Line items:
Line 18: Spell out KB, do not use abbreviation here.
Line 19: Please check: count should be counted
Line 21: Please review the sentence, the word data is awkward and should be removed.
Keywords suggestions: All words in the title are key words. Suggest removing Acidovorax citrulli and competition from the key words and add new ones that are not in the title.
Introduction:
Line 51: consider revising to: “ what is the explanation for this phenomenon?”
Line 61: consider revising: I don’t think its instar specific detection assays. Is it inter-specific detection assays ?
Material and Methods:
Line 81: Here the authors define melon and watermelon with the scientific name. They may want to do this at first mention in the introduction.
Lines 87, 93, 169,183, ect.: Please spell out the first word of a sentence: Acidovarax
Lines 100, 128, 130, ect.: Please be consistent with the spacing of the units. For example: 5mL or 5mL
Line 113: Suggest not starting the sentence with pslb65. Perhaps: The strain pslb65…
Line 114 and line 106: Be consistent. Ten microliters
Line 160: was used as…: this sentence is awkward as written. Perhaps it’s: was used as the ordinate to generate the standard curve.
Lines 176-177: I don’t believe there was an indication on how long the bacteria was allowed to infect. Only says sampled each day. Can the author state how long they monitored.
Line 184: Spell out, do not abbreviate the first word of a sentence.
Line 197: Unsure what AUPGC is in parentheses.
Results: There is some interpretation in the results. Please move these sentences or ideas to the discussion.
For example: Line 221-224: Inhibition activity assays revealed that the ……
This is interpretation. Just state that there was not significant inhibition. Avoid using words like revealed and cannot. Just state the result and interpret the result in the discussion.
Lines 227-239: Does this section go with the figure legend? If so, format a little clearer.
Line 252: there is no need to have (AUPGC) since it is spelled out. Please adjust.
Line 260: review spacing of the units (18h or 18 h). Be consistent.
Line 289: Please change stans to strains.
Line 313-314: this could be interpretation. Just state the first part and expand on why there might be differences in the discussion.
Discussion: See my first comment above.
Line 403: separate the words thosereported to: those reported
Line 406: Please check but perhaps change show to showed
Line 409-410: I am not sure the authors present enough information to support this conclusion. I would suggestion adding.
Line 415: This is a little confusing because the author discusses lots of genetic information based on spray inoculation, which is not the same as injecting it into the cotyledons. Injecting may cause a whole bunch of other host responses so all this information should be put into context.
Line 417: This is where HAI should be defined, not in the abstract.
Line 429: Be consistent, h after-inoculation should be HAI
In general, there are minor issues throughout but the results from this study need to be discussed in more detail in the discussion section. Please remove any interpretation from the results and discuss them in the discussion section. The authors can add the appropriate genetic possible reasons, but should be brief.
Reviewer 2 Report
The manuscript in the present form needs many improvment specially the results. All figures did not show letters that corresponding to significative differences among treatments. References are ok, as well as Introduction, MM and discussion. Title of X and Y axis muste be in capital letter.
Round 2
Reviewer 1 Report
Further characterization of host preference of Acidovorax citrulli based on growth competition between group I and group II strains
My major concerns have been addressed and the manuscript is much clearer. Below are some minor corrections.
Line 86: need space before the °C for both
Line 91: srtrain needs to be strain
Line 102: 6000 r/min need to be converted to cf or g because RPM is different across machines. Therefore, for reproducibility please convert.
Line 116,118,120,125,131,132,153: all need a space before the °C
Line 196: Please make section 2.6 all one paragraph. The first paragraph is only one sentence.
Line 227: add a space before h (12h) should be (12 h)
Line 230: no need to spell out hours. Should be h
Line 233: delete the extra period.
Line 252: please replace suggest with indicated
254: add a space before h
Line 258-259: if the values are p values, please label as such…. Example (…was P=0.120 ….and was P=0.555). Also, needs to be in the past tense so is should be was.
Line 264: need space before the °C
Line 282: Is there an extra space between strains population?
Line 386: Please change to (P > 0.05). The spaces are missing to make it consistent with the others.
Line 432: Need to change to group I and group II. The spaces are missing to make it consistent with the others.
Line 469: srains needs to be changed to strains
